# Interpretability of Spatiotemporal Dynamics of the Brain Processes Followed by Mindfulness Intervention in a Brain-Inspired Spiking Neural Network Architecture

**DOI:** 10.3390/s20247354

**Published:** 2020-12-21

**Authors:** Zohreh Doborjeh, Maryam Doborjeh, Mark Crook-Rumsey, Tamasin Taylor, Grace Y. Wang, David Moreau, Christian Krägeloh, Wendy Wrapson, Richard J. Siegert, Nikola Kasabov, Grant Searchfield, Alexander Sumich

**Affiliations:** 1Faculty of Medical and Health Sciences, School of Population Health, Section of Audiology, The University of Auckland, Auckland 1142, New Zealand; g.searchfield@auckland.ac.nz; 2Eisdell Moore Centre, The University of Auckland, Auckland 1142, New Zealand; 3Centre for Brain Research, The University of Auckland, Auckland 1142, New Zealand; d.moreau@auckland.ac.nz; 4Information Technology and Software Engineering Department, Auckland University of Technology, Auckland 1010, New Zealand; mgholami@aut.ac.nz; 5School of Psychology, Nottingham Trent University, Nottingham NG25 0QF, UK; mark.crookrumsey@ntu.ac.uk (M.C.-R.); alexander.sumich@ntu.ac.uk (A.S.); 6Faculty of Medical and Health Sciences, The University of Auckland, Auckland 1142, New Zealand; t.taylor@auckland.ac.nz; 7Department of Psychology and Neuroscience, Auckland University of Technology, Auckland 0627, New Zealand; grace.wang@aut.ac.nz (G.Y.W.); chris.krageloh@aut.ac.nz (C.K.); richard.siegert@aut.ac.nz (R.J.S.); 8School of Psychology, The University of Auckland, Auckland 1142, New Zealand; 9School of Public Health and Interdisciplinary Studies, Auckland University of Technology, Auckland 0627, New Zealand; wendy.wrapson@aut.ac.nz; 10Intelligent Systems Research Centre, Ulster University, Londonderry BT48 7JL, UK; 11School of Engineering, Computing and Mathematical Sciences, Auckland University of Technology, Auckland 1010, New Zealand

**Keywords:** mindfulness, oddball-paradigm event-related potential (ERP) data, target and distractor stimuli, dynamic spatiotemporal brain data, computational modelling, spiking neural network

## Abstract

Mindfulness training is associated with improvements in psychological wellbeing and cognition, yet the specific underlying neurophysiological mechanisms underpinning these changes are uncertain. This study uses a novel brain-inspired artificial neural network to investigate the effect of mindfulness training on electroencephalographic function. Participants completed a 4-tone auditory oddball task (that included targets and physically similar distractors) at three assessment time points. In Group A (*n* = 10), these tasks were given immediately prior to 6-week mindfulness training, immediately after training and at a 3-week follow-up; in Group B (*n* = 10), these were during an intervention waitlist period (3 weeks prior to training), pre-mindfulness training and post-mindfulness training. Using a spiking neural network (SNN) model, we evaluated concurrent neural patterns generated across space and time from features of electroencephalographic data capturing the neural dynamics associated with the event-related potential (ERP). This technique capitalises on the temporal dynamics of the shifts in polarity throughout the ERP and spatially across electrodes. Findings support anteriorisation of connection weights in response to distractors relative to target stimuli. Right frontal connection weights to distractors were associated with trait mindfulness (positively) and depression (inversely). Moreover, mindfulness training was associated with an increase in connection weights to targets (bilateral frontal, left frontocentral, and temporal regions only) and distractors. SNN models were superior to other machine learning methods in the classification of brain states as a function of mindfulness training. Findings suggest SNN models can provide useful information that differentiates brain states based on distinct task demands and stimuli, as well as changes in brain states as a function of psychological intervention.

## 1. Introduction

### 1.1. Mindfulness Training: The Neural Basis for Performance Monitoring

Mindfulness, derived from Buddhist traditions and adapted to Western secular contexts, is supported as an effective component in interventions for mood disorders (e.g., anxiety, depression, stress [1,2,3,4]), and sensory disorder (e.g., tinnitus [5]) and may influence several implicated neurocognitive domains (e.g., executive function, attention, emotional regulation, inhibition, and awareness) [6,7]. Improved cognitive performance following mindfulness training has been demonstrated using the Stroop [8,9,10], attentional switching [11], and sustained attention [12] tasks. Additionally, behavioural performance on the Go/No-go test (measuring sustained attention and inhibitory control) has been reported after a 3-month mindfulness retreat to reliably predict improved socioemotional function for up to 5 months [13].

Underpinning mechanisms of these changes have been investigated using neuroimaging [14,15] and electroencephalography (EEG) methods [16,17,18,19,20]. For example, alterations in resting-state EEG parallel improvement in mood following mindfulness intervention [21]. That is, normalisation of connectivity weights at frontal and temporal sites was seen following mindfulness training, particularly in a group that had the greatest reduction in depression. Resting-state functional connectivity changes with mindfulness practice have also been observed between the dorsolateral prefrontal cortex (DLPFC) and default mode network areas (e.g., posterior cingulate cortex and dorsal anterior cingulate [22,23]), in line with findings of decreased mind-wandering and increased capacity for attention shifting in experienced meditators [15,24].

Further understanding of the functional significance (e.g., cognitive control and attention) of such changes in frontotemporal function is possible through the evaluation of task-related EEG activity, such as that measured by event-related potentials (ERPs). ERPs provide a direct measure of sensory/perceptual (before 200 ms e.g., P1, N100, P150) and higher-level (after 200 ms e.g., N200, P300) cognitive operations associated with stimulus presentation and/or task performance [25,26]. For example, previous work involving ERPs has reported that meditation can enhance the facility to efficiently and rapidly allocate and reallocate attention, with alterations seen at early information processing stages [27,28,29,30].

Other studies report mindfulness-associated alterations in later stages in information processing [31,32,33,34]. Two ERP components commonly investigated during ERP Go/No-go tasks (indexing cognitive control) include the N200 and P300. N200 occurs around 180–250 ms post-stimulus and has been associated with cognitive control, response inhibition, and conflict monitoring. It typically shows higher amplitude in No-Go compared to Go trials [35,36]. The P300 ERP component and late positive potential (LPP) have been proposed to reflect attentional, executive, and cognitive control processes [37]. Typically the P300 is observed between 280 and 400 ms after the presentation of the stimulus to salient and/or task-relevant stimuli [27,37]. It is sensitive to a wide range of stimulus characteristics and cognitive processes, ranging from task difficulty, stimulus category, and resource allocation [38]. N200 and P300 are composite waveforms comprising several subcomponents that vary in topography and latency. For example, subcomponents of the P300 include earlier anterior (P3a) responses to novelty and/or salience [39,40,41], mid-range posteriorly distributed (P3b) components associated with working-memory updating and target detection [41,42] and later anterior (P3c) responses, reflecting inhibition and behavioural modulation [43,44,45]. Subcomponents differentially respond to task demands. For example, P300 amplitude is generally higher and anteriorly distributed in No-Go, relative to GO trials and is related to the assessment of response inhibition in those trials [43,45].

The effect of meditation on ERPs depends on the type of mediation, processing stage investigated, and eliciting task used [31,46]. Some studies report increases in N200 amplitude [47], whilst both increases and decreases in P300 have been reported. Cahn and Polich [46], for example, suggest P3a (along with P200 and N100) amplitudes in response to white noise bursts is reduced following Vipasana meditation, whilst P3b may be increased. Others also report increased P3b with meditation (for review see Cahn and Polich, 2009). Moore et al. [47] report decreased P300 (centro-parietal sites alone were reported) to incongruent stimuli on a colour-word Stroop task in meditators compared with a nonmeditators group. In comparison, higher P300 amplitude was seen at frontocentral brain regions in response to target stimuli (frontal P3b) following meditation practice, suggesting an improvement in frontal attention that was related to subsequent memory processing [41,48,49]. Additionally, mind-wandering, which is often reduced during mindfulness, covaries with a reduction in P3b [50].

In a functional magnetic resonance imaging (fMRI) study, meditators and nonmeditators of performance were assessed on impulse and attention control using a Stroop Task [51]. In comparison to meditators, nonmeditators demonstrated increased activity in the temporocentral, frontocentral, pre and postcentral gyri, and basal ganglia during incongruent conditions. These outcomes from neuroimaging studies suggest that meditation may improve efficiency, by enhancing the ability to control impulses [51] and sustain attention [52,53,54].

### 1.2. Computational Modelling of Data in a Brain-Inspired Spiking Neural Network Architecture

Recent studies have begun to use machine learning methods to integrate the dynamic patterns of spatiotemporal brain data contained in EEG and ERP data [55,56,57,58,59,60]. This poses a challenge, as the temporal component may display complicated interactions that dynamically alter over time [61]. Thus, spatial and temporal information is often separated for analysis, rather than incorporating these aspects into one unifying computational model. Such a model should be constrained by “biological plausibility” [62,63] (inspired by the function of neurological systems—Definition in Appendix A (1)) and be interpretable. The model’s interpretability should facilitate comprehension of why certain outcomes have resulted from the model, and discoverability of causal interactions that have resulted in the output. To this end, the current research applies a novel computational framework based on one of the most promising trends of artificial neural networks (ANN), called spiking neural networks (SNN). SNN models have been developed with a neurobiologically-plausible computational architecture that incorporates both spatial and temporal characteristics of data into one unifying model and can be applied for pattern recognition. Classification, and prediction of different mental and neurological cognitive states [64,65,66]. They are considered a suitable tool for the analysis of the spatiotemporal brain data (STBD), where both space and time components are crucial to be learnt [57,58,67,68,69,70].

The current research applies SNN to ERP data in order to investigate brain mechanisms of cognitive control in relation depression and mindfulness. It addresses the following aims to:(i)design a SNN architecture to model and understand brain activity patterns generated before and after mindfulness training for spatiotemporal relationships between continuous ERP data streams, while participants of two groups (experimental and waitlist) were responding to (1): target stimuli (2) and nontarget (distractor) stimuli.(ii)create computational models from ERP data to better scrutinize the temporal dynamic processes at different time points of stimulus presentation.

## 2. Materials and Methods

### 2.1. Ethics

Ethical approval was obtained from the Auckland University of Technology Ethics Committee (AUTEC), New Zealand. All experiments were performed under AUTEC guidelines and regulations. Before the assessment, each participant was fully advised of their rights and provided written informed consent. Data were collected at the Faculty of Health and Environmental Sciences, Auckland University of Technology, New Zealand.

### 2.2. Data and Software Availability

The software used for the implementation of the designed method can be found at http://www.kedri.aut.ac.nz/neucube/. Whilst current ethical approval does not allow us to make data available, the authors are currently applying to make data available in future.

### 2.3. Participants

Forty participants were recruited as part of a mindfulness intervention study [71] and underwent baseline (T1) assessments that included EEG/ERP, psychometrics (Beck’s Depression Inventory (BDI) [72], Depression Anxiety Stress Scale (DASS) [73], and Five Facet Mindfulness Questionnaire (FFMQ) [74]), and blood measures [75] (not reported in the current study) (the definitions of the tests are in Appendix A (2), (3), (4)). Twenty people were selected for this study based on (i) having completed EEG assessments at all time-points (see below); (ii) shown an increase in mindfulness scores and (iii) the absence of severe depression. Of these *n* = 20, one group (experimental, *n* = 10; 3 male and 7 females; age = 19–57 years; mean = 29, SD = 10.58) had been enrolled into the mindfulness training programme following baseline assessments (T1) and were then assessed at 6 weeks following the mindfulness training programme (T2,) and 6 weeks follow-up after T2 (T3). Another group (waitlist, *n* = 10, 2 male and 8 females; age = 18–56 years; mean = 28.80, SD = 9.74) had been enrolled onto a waitlist (T1-T2, 6 weeks), and then underwent mindfulness training following which (6 weeks) they had a third assessment (T3).

### 2.4. Trait Mindfulness 

Changes in mindfulness were calculated as the difference between post- and pre-training by the FFMQ [76,77] score. Table A1 of the Appendix B provides descriptive information of all participants and their FFMQ (Post-Pre) scores.

### 2.5. Mindfulness Intervention

The mindfulness intervention was revised from an educational program called “Pause, Breathe and Smile” [78]. Each mindfulness phase took 90–110 min and consisted of discussion and regular brief guided meditation exercises. A full explanation of the programme has been summarised in the previous research [71]. Figure 1a–e show the data collection protocol and SNN-based methodology for Spatiotemporal Brain Data (STBD) modelling, learning, visualisation, comparison, and classification. EEG data were recorded at three time points in each group (Figure 1a,b). In the experimental group, these included assessments at baseline, time point 1 (T1, PRE1), post 6 weeks mindfulness training (T2, POST1) and follow-up (T3, POST2). In the waitlist group, assessments were conducted at baseline (T1, PRE0), following 6 weeks on a waitlist (T2, PRE1) and following 6 weeks mindfulness training (T3, POST1), (shown in Figure 1b).

### 2.6. ERP Cognitive Task

Participants completed a 4-tone auditory oddball task undertaken in a quiet room. Stimuli were randomly presented through headphones (64 dB) across two blocks, each containing 90 standard tones (1000 Hz, sine, duration = 40 ms, attack/decay = 5 ms), 18 white noise bursts (duration = 40 ms), 18 targets, and 18 distractors. Thus, there were 288 stimuli in total with noise bursts, targets, and distractors each comprising 12.5% of stimuli. This is necessarily lower than the probability of target stimuli in a typical 2-tone oddball (often around 0.2), in order to balance requirements for maintaining a high frequency of standard tones (62.5% of trials in the current study), generate sufficient trials for all stimuli, and minimise duration of the whole task. Similarly, probabilities in Cahn and Polich (2009) were 0.10 for targets and white noise bursts. Targets and distractors were 1500 Hz sine wave tones attack/decay = 5 ms. Distractors were assigned the same frequency as targets in order to generate cognitive conflict, requiring activation of cognitive control and inhibitory mechanisms (P3c) in the case of the distractors which differed from targets on a second feature, duration. In block A, targets were 40 ms and distractors were 100 ms tones. In block B targets were 100 ms tones and distractors were 40 ms tones. The block order was randomly counterbalanced across participants. The noise bursts, which were designed to activate an orienting response (P3a), are not included in the current study. In all cases the interstimulus interval was 1100 with 100 ms jitter. 

Participants were asked to click on a button with their right (dominant) hand as fast and accurately as possible once they heard the target sound, and to refrain from pressing any buttons to any other sound. Prior to commencing data collection, participants were presented three times with each stimulus and allowed sufficient practice to demonstrate that they understood instructions and could discriminate the target from distractor stimuli.

### 2.7. EEG Acquisition and Pre-Processing

During the task, a Synamps amplifier (NeuroScan; sampling rate = 1000 Hz; AFz ground) and quikcap (based on the 10–20 system) were used to record EEG data from 62 channels: FP1, FPZ, FP2, AF3, AF4, F7, F5, F3, F1, FZ, F2, F4, F6, F8, FT7, FC5, FC3, FC1, FCZ, FC2, FC4, FC6, FT8, T7, C5, C3, C1, CZ, C2, C4, C6, T8, TP7, CP5, CP3, CP1, CPZ, CP2, CP4, CP6, TP8, P7, P5, P3, P1, PZ, P2, P4, P6, P8, PO7, PO5, PO3, POZ, PO4, PO6, PO8, CB1, O1, OZ, O2, CB2. Impedance was maintained below 10 kOhms. Curry 7.12 software was used to pre-process data off-line. The signal was re-referenced to linked A1/A2, corrected to baseline, and subject artefact removal. Initially, vertical ocular artefacts (e.g., blinks) were reduced by identifying activity in EOG and midline frontal electrodes that exceeded threshold (60 micoVolts). Principal components analysis was performed on data surrounding (−200–500 ms) the artefact peak to reduce the artefact. This was repeated using bilateral frontal electrodes to reduce any residual artefact. Data was then reduced to epochs of −100–900 ms surrounding stimulus onset. Epochs were scanned manually for excessive EMG artefact and if present the Epoch was removed from analysis. 

The current study reports on target and distractor stimuli across 20 participants in two groups (experimental and waitlist) over three time points (T1, T2, and T3). Data collected at T1 and T2 was used for visualisation; T1, T2, and T3 for classification. The data was averaged to form ERPs for target and distractor stimuli. Twenty samples were created (10 for experimental and 10 for the waitlist group), each sample file contains data of one participant (1300 datapoints which represent the length of ERP epoch).

### 2.8. Method: Brain-Inspired Spiking Neural Networks

SNN models are neuro-computational units that are stimulated with respect to neural structure in the brain. According to several authors [70,79], the SNN follows a biologically plausible structure, in that:-A brain template is utilised to build a 3-dimensional space of spiking neurons (SNN model) that reserve spatial information of brain structure. Mapping of brain data variables into the SNN model.-Input spatiotemporal data (brain signals) are converted to a ‘spike train’ (series of binary events when the brain signal reaches a threshold value).-The SNN model is initialised using the brain-inspired small-world connectivity rule.-The initialised SNN model learns through a biologically plausible learning algorithm to adapt the model’s connectivity, resulting in long chains of connections.

In a SNN model, every “neuron” (more definition is in Appendix A (5)) is an information processing element, facilitated with a learning algorithm that extracts the relationship between the steaming data variables over time. That is, spiking neurons are connected by synapses, where the learning patterns are memorised. Compared with conventional machine learning methods [80,81,82], SNN models integrate the notion of time into the computation and thus are considered to be superior in biological plausibility in neural networks compared to previous models that do not account for temporal dynamics. Thus, SNNs are recognised as appropriate models for processing STBD [79].

In this study, the SNN connectivity captured the relationships between the ERP variables in a computational model rather than the precise organisation of the brain’s physical neural connectivity. The SNN architecture includes several modules: a data encoding procedure; a 3D SNN model that learns from data in an unsupervised mode; a layer of spiking neurons for supervised learning; output classification; optimisation; finally, interpretation of SNN models and knowledge extraction [79,83]. Figure 1c illustrates the designed methodology, containing segmentation of data based on the averaged ERP time points and ERP encoding into spike sequences; while Figure 1d,e present the computational modelling of data into a 3D space of artificial neurons, patterns recognition, and patterns classification.

In the following the steps for computational modelling of data will be explained.

#### 2.8.1. Initialisation and Encoding of the ERP Data in the SNN Model

In the first phase of data modelling, the real-value ERP time series need to be encoded to train of spikes using an appropriate encoding method. The encoded spike trains reflect the significant changes in data over time. In this case, the Threshold-Based Representation (TBR) [84] was used as the encoding method. Using this technique, if the upward change in a signal’s amplitude is more than a threshold at a certain time, then a positive spike is produced. Conversely, negative spikes are created if the amplitude diminutions below the defined threshold. When none of these cases, then no spikes are generated.

The generated spike trains embody changes in the STBD that exceeded a threshold TBRthr. Figure 1c demonstrates an instance of encoded EEG signals into positive and negative spike trains generated from the raw EEG data. Figure 2 and Figure 3 depict the activated areas (based on the spiking activity) in the SNN models of target and distractor stimuli at different ERP latencies (253 and 313 ms post-stimulus) for an experimental participant at two time points (T1, T2). This confirmed the clear discrimination between the brain responses to target and distractor stimuli before and after the mindfulness training. 

Figure 2a,b show that at T1 more spikes were activated around frontal, centroparietal, and occipitoparietal at 253 and 313 ms post target stimulus. However, after the mindfulness training (Figure 2c,d), a greater number of spikes were triggered around frontal and temporal areas compared with T1.

Figure 3 demonstrates that when participants responded to the distractor stimuli, the size of activated areas in SNN models was greater than the target stimuli across the scalp. Figure 3a,b show that more clusters of spikes activated at 313 ms post-stimulus across the scalp than 253 ms. The generated clusters were stronger (involved more activated neurons shown in red) around the frontal region at T2 (Figure 3c,d).

After spiking activity in the model, a model was then prestructured to represent the functional and structural information of the brain processes measured by spatiotemporal data (Figure 1d). The STBD data samples were mapped spatially into 3D artificial neural space where the spatial information of brain areas is topologically preserved concerning the (x, y, z) coordinates as positioned in brain Talairach atlas [85]. In the SNN model, after defining a biologically plausible 3D SNN, data were initialised with a Small-World Connectivity rule (SWC) [86] that defines a probability by which a neuron *i* can be linked to a neuron *j* with respect to their internal distance, the greater the distance between i and j the smaller in the connection probability. The generated initial connections were adapted during the unsupervised learning process which takes into account the temporal dynamics of input data (described in the following section).

#### 2.8.2. Unsupervised Learning Mechanism in a SNN Model

The SNN model uses an unsupervised learning algorithm, called Spike Time-Dependent Plasticity (STDP) which allows the model to learn the spatiotemporal relationships in the input spikes [79,87,88]. This learning process modifies the neural connection strength with respect to the timing of pre to postsynaptic neurons. The connections between neurons were updated dynamically at each time point of the input data (e.g., at a millisecond scale), resulting in deep trajectories of connectivity learnt in the 3D SNN structure. Throughout the STDP learning process, if every neuron’s potential passes an activation threshold in time t, then it produces an output spike. The spike is then transferred to other neurons linked to it. This neuron likewise keeps receiving spikes over time and, after passing a threshold, fires [89]. In this way, spikes are propagated inside the SNN model during the STDP learning and the ‘hidden’ spatiotemporal relationships between the data variables are captured in the shape of neural connectivity.

#### 2.8.3. Supervised Learning and Pattern Classification in SNN Models

An output layer of spiking neurons is used and trained in a supervised mode to classify the spiking activity of the model identified with various stimuli (e.g., target, distractor) related to “before” and “after” the mindfulness training (two classes) to compare the two SNN models and to interpret the differences (Figure 1e).

An output classifier layer of dynamic evolving SNN (deSNN) [69,79,84] is built based on supervised learning to learn the relationship between the EEG data and their class labels. This classifier is evolving and allowing for new data to be introduced to the model and tested incrementally. Here, for every EEG sample (from the training dataset), one neuron is generated and positioned on the output layer which is fully connected to the whole SNN space. These initial connections between the SNN model and the output layer are initialised using rank-order rule and then modified when the input data were again passed into the model for supervised learning. In the validation phase, the EEG samples which were excluded from the training are used as testing data. The classification here is based on the k-nearest neighbours (kNN) method applied to the distance between the newly generated output neurons (testing samples) and that of any one of the already trained output neurons. When the output of a new testing sample was not known in advance, the class label of this sample was defined by the label of the majority of the output neurons [55,70] selected by the kNN method.

#### 2.8.4. SNN Model Evaluation

To evaluate the level of significance in the trained SNN models, each experiment was performed with respect to the following criteria: for a mental activity (called class) of each individual data, one SNN model is created. Each class of data contains *n* samples that are used to train the SNN model through an iterative procedure of leave-one-out as follows (shown in Figure 4): 

The SNN model is initialised.The initialised SNN model is trained with (*n* − 1) samples (one sample is excluded from the training).The average of the quantitative information (spatiotemporal connectivity) in the trained SNN model is calculated.

The hold-out sample is replaced by another sample, then it returns to step 1 until all the samples are excluded from the training set, one by one. This means that a set of SNN models are initialised and trained with different folds of samples.

The numerical information of the trained SNN models can be also statistically examined to evaluate the models’ significance. To this end, for every trained SNN model, an activation level was measured through computing the average value of its connection weights (definition is in Appendix A (6)). Here, we applied ANOVA (analysis of variance) to the activation levels of several trained SNN models to evaluate the significance of the models. 

## 3. Results

### 3.1. Pattern Recognition, Visualisation, and Mapping

The results from the designed SNN-based methodology (that was fully explained in the methods section), is depicted graphically in Figure 1 and consists of the following steps: (1)Mapping, modelling, learning, classifying, and understanding of ERP data.(2)Statistical and quantitative analysis was performed on the SNN models to assess the model significance.

Firstly, a brain-inspired 3D SNN model was designed based on the Talairach brain atlas of 1471 neurons. Here, the term “neuron” is used to represent the centre co-ordinate of one cubic centimetre area from the 3D Talairach Atlas [85]. The SNN model input neurons are allocated to the 62 EEG channels to transfer their spike trains into the SNN model. 

EEG channels were divided into five regions for their anatomical component (Figure 5): frontal, frontocentral, temporal, centroparietal, and occipitoparietal. Then, eight separate 3D models were trained with different ERP datasets related to target and distractor stimuli across experimental (T1, baseline; T2, post-mindfulness training) and waitlist (T1 baseline; T2, post-waitlist lead-in) groups (Figure 6, Figure 7, Figure 8 and Figure 9). Figure 6 and Figure 7 show the neuronal connections created for the experimental group in the brain-inspired SNN models reflect the functional connectivity in response to target (Figure 6) and distractor stimuli (Figure 7) before and after the mindfulness training; while Figure 8 and Figure 9 represent the neuronal connections created for the waitlist group in the brain-inspired SNN models reflect the functional connectivity in response to target (Figure 8) and distractor stimuli (Figure 9) before and after the mindfulness training.

As shown in Figure 6b, after the mindfulness training, stronger connectivity was generated around the frontal regions in the experimental group, compared with the waitlist group in response to target stimuli (Figure 7b). Distractor stimuli showed lower connectivity at T1 compared to T2 for both groups (Figure 7a and Figure 9b). 

The differences between the SNN models of T1 and T2 sessions can be also studied by computing the amount of spatiotemporal network interactions between the EEG variables using a feature interaction network (FIN). In Figure A1 of the Appendix B, the total temporal spike interaction among 62 input neuronal areas (corresponding to 62 EEG channels) for experimental group towards target stimuli before and after the mindfulness training is shown in the FIN, where nodes represent the input neuronal areas (neuronal clusters) and each line, that links two nodes, corresponds to the amount of spike transmission between the clusters during the SNN learning model.

The numerical information of the trained SNN models can be statistically analysed to evaluate the models’ significance. To this end, for every trained SNN model, an activation level was measured through computing the average value of its connection weights.

Therefore, for every participant, one SNN model was developed at T1 and T2. The average connection weights for each developed individual SNN model were calculated as a function of group (experimental, waitlist) and time (T1, T2). The averaged connection weights of each developed SNN model for each group in response to target and distractor stimuli across all time of data collection are reported in Table 1 and were used for further statistical analysis.

### 3.2. Statistical Analysis of the SNN Models

The connection weights for each EEG channel were then divided into five sites for each hemisphere with their topographical features (as shown in Figure 5): frontal, temporal, frontocentral, centroparietal, and occipitoparietal. A repeated measures analysis of variance (ANOVA) was performed to assess differences in functional activity between time and stimuli types (Table 2). Independent variables include *Time* (T1, T2), *Hemisphere* (left, right), *Site* (centroparietal, occipitoparietal, frontal, temporal, and frontocentral), *Stimuli* (target, distractor), and *Group* (experimental, waitlist). All violations of the assumption of sphericity were corrected using Greenhouse–Geisser corrections.

The distribution of the connection weights can be seen in Figure 10 as a function of *Site*, *Group,* and *Time*. The SNN connection weights distribution related to “before” mindfulness (green colour) and six weekly mindfulness follow-ups (pink colour) across five sites of the brain in experimental and waitlist groups in response to target stimuli (Figure 10a,c) and distractor stimuli (Figure 10b,d) across the whole EEG epoch.

Firstly, the weights were examined at T1 in order to investigate differences between the stimuli pre-mindfulness training. There were significant *Site* * *Stimuli* [F (4, 72) = 85.225, *p* < 0.001, ηp^2^ = 0.826] and *Site* * *Hemisphere* * *Stimuli* [F (2.572, 46.302) = 13.254, *p* < 0.001, ηp2 = 0.424] interactions. These were due to lower weights over frontal [F (1, 19) = 129.832, *p* < 0.001, ηp^2^ = 0.872], and greater weights over parietooccipital [F (1, 19) = 63.047, *p* < 0.001, ηp^2^ = 0.768] sites bilaterally for targets compared to distractors. In addition, in the left hemisphere only, there were higher weights for distractors than targets at temporal [F (1, 19) = 37.291, *p* < 0.001, ηp^2^ = 0.662] and frontocentral [F (1, 19) = 22.678, *p* < 0.001, ηp^2^ = 0.544] sites. No differences were found between target and distractor stimuli over centroparietal regions [F (1, 19) = 0.419, *p* = 0.525, ηp^2^ = 0.022].

Separate ANOVAs were conducted for target and distractor stimuli to investigate the effects of mindfulness training. Within groups variables included *Time* and *Site* with *Group* as a between groups variable. These analyses only considered effects that included *Time* and *Group* interactions.

For target stimuli, there was a significant *Site* * *Time* * *Group* interaction [F (2.220, 39.956) = 61.419, *p* = 0.002, ηp^2^ = 0.204], which appeared due to a significant *Time* * *Group* interaction at frontal sites [F (1, 18) = 9.853, *p* = 0.006, ηp^2^ = 0.354]. Although there was an increase in weights between T1 and T2 for both experimental [F (1, 9) = 66.066, *p* < 0.001, ηp^2^ = 0.880] and waitlist [F (1, 9) = 13.577, *p* = 0.005, ηp^2^ = 0.601] groups, the effect was greater for the experimental group. Thus, whilst no difference was seen between groups at T1 [F (1, 18) = 0.048, *p* = 0.829, ηp^2^ = 0.003], at T2 the experimental group had greater weights than the waitlist group [F (1, 18) = 9.826, *p* = 0.006, ηp^2^ = 0.353]. For the distractor stimuli, there was a significant *Group * Time* interaction [F (2.884, 51.918) = 4.599, *p* = 0.007, ηp^2^ = 0.204], and this was due to a significant effect of *Time* (T2 > T1) in the experimental group [F (1, 9) = 40.843, *p* < 0.001, ηp^2^ = 0.819], which was not seen in the waitlist group [F (1, 9) = 1.112, *p* = 0.317, ηp^2^ = 0.111].

### 3.3. Correlations between Connections Weights, Mindfulness, and Psychometrics Data

A Kendall’s Tau correlation was computed to assess the relationship between the created connection weights in the SNN model across brain regions for both target and distractor stimuli at baseline (T1) and the psychometric scores including (BDI) [90], (DASS) [91], and the (FFMQ) [74].

Kendall’s Tau correlations showed right frontal connection weights were significantly positively associated with FFMQ (target *rτ* = 0.378, *p* = 0.033; distractor *rτ* = 0.43, *p* = 0.015). The response to distractors was also inversely associated with Depression as measured by the DASS (*rτ* = −515, *p* = 0.005) and BDI (*rτ* = −0.508, *p* = 0.004). Associations between right frontal connection weights in response to distractors and other DASS measures were on the threshold for significance (Anxiety *rτ* = −0.363, *p* = 0.05; Stress *rτ* = −0.348, *p =* 0.06).

### 3.4. Pattern Classification and Discrimination

After training the SNN models, a classifier was trained to classify the SNN model activity for the participants’ responses towards target stimuli at T1, T2, and T3, across two participant groups. In total, 30 samples were created and divided (10 samples per class = number of participants in each class) into three classes (class 1: response to target stimuli at T1; class 2: response to target stimuli at T2; class 3: response to target stimuli at T3). The same division of samples for the waitlist group was defined. Table 3 represents the classification results for the three classes. To perform a comparative analysis between the performance of the used dynamic evolving SNN method (deSNN) [69], and traditional machine learning methods, we used multilayer perceptron (MLP) [81,92], multiple linear regression (MLR) [82], and support vector machine (SVM) [63,93] for classification of EEG data as reported in Table 3. The classification procedure was performed using leave-one-out cross-validation. Table 3 reports the accuracy of classification resulted from the SNN models versus the conventional machine learning techniques. The confusion table here demonstrates the number of misclassified and correctly classified EEG samples.

## 4. Discussion

This is the first study to apply a brain-inspired SNN model, incorporating temporal and spatial components of EEG data, to investigate changes in brain function associated as a function of mindfulness training. 

Several key findings have emerged from this work. In terms of using a SNN model to differentiate the EEG response to target compared to distractor stimuli at baseline, the connection weights were lower for targets than distractors over bilateral frontal, left frontocentral, and left temporal regions, but were greater for targets over left occipitoparietal regions. Right frontal connection weights in response to distractor stimuli at baseline were associated positively with FFMQ scores and inversely with Depression. Following mindfulness training, relative to the waitlist group, the experimental group showed a greater increase in frontal connection weights in response to targets. Thus, by follow-up, frontal connection weights were greater in the experimental group than the waitlist group. In response to distractors, a general increase across sites was seen in connection weights following mindfulness training in the experimental group, whilst the waitlist group showed no difference between T1 and T2. Finally, the accuracy of spatiotemporal data classification was higher when using SNN models compared with traditional machine learning methods (e.g., SVM, MLP, and MLR).

Previously, we applied SNN to resting-state EEG data to differentiate states of depression and the effect of mindfulness training on brain function [56]. That study reported mindfulness training increased resting-state connection weights, in parallel with improvements in depression; and that baseline connection weights predicted response to mindfulness training. Another study from our group applied SNN to ERP data in response to familiar and unfamiliar logos [57].

Traditionally, ERPs are measured according to amplitude and latency [27]. The current study applied a novel technique that capitalises on the temporal dynamics of the shifts in polarity throughout the ERP, spatially across electrodes, as inputs to the SNN (depending on the threshold). The SNN is then used to model the connectivity weights with respect to the (STDP) learning rule [87]. According to STDP rule, connectivity with greater weights reflects stronger spike transmission between neurons. Thus, the STDP rule develops neural connectivity among the spatially distributed neurons in the SNN model that represents the spatiotemporal relationships between the brain data variables.

### 4.1. Comparison of Stimuli (Target vs. Distractor)

Our findings are consistent with previous ERP amplitude research that has shown a more posterior maximum in response to target stimuli (e.g., P300), which may be underpinned by activity in several distributed regions including medial and lateral prefrontal regions, temporoparietal junction, and inferior parietal lobule [94,95]. In comparison, ERP anteriorisation is seen in response to distracting, stop and/or No-Go stimuli [43,96]. In the current study, this was supported by the bilateral increase in frontal connection weights in response to distractor (relative to the target) stimuli. Greater activation weights to distractors over left frontocentral and temporal sites might reflect the activity of the DLPFC which, along with the ventrolateral prefrontal cortex (VLPFC), superior parietal cortex, and frontal eye field has been associated with a distractor network [97,98].

A similar network involving the DLPFC, orbitofrontal cortex, dorsal anterior cingulate cortex, and amygdala is proposed to underpin deficits in cognitive control in depression [99]. Our current findings from the correlation analysis support this idea, at least for right frontal connection weights in response to distractor stimuli, which were associated with both depression and mindfulness and may reflect mechanisms involved in interference control. Thus, right frontal networks of cognitive control may represent a mechanism by which mindfulness training reduces depression. Further research will however need to extend these findings from the general population to clinical populations and investigate the clinical utility of applying SNN models to ERP data. For instance, in predicting response to interventions, as has been shown for resting-state data [56].

### 4.2. Effect of Mindfulness Training (T1 vs. T2)

As shown in Figure A1 of the Appendix B, broader interaction lines were formed between the 62 EEG data variables of target stimuli at T2 compared to the T1. In the case of the target stimuli, there were thicker interaction lines in the SNN model at T2, especially between the EEG channels positioned at frontal, frontocentral, and temporal areas, when compared with the models at time T1 (before training). The current findings suggest that mindfulness training results in an increase in connectivity, particularly in relation to frontal function and executive control. Previous studies have shown improved behavioural performance during cognitive distraction (e.g., Stroop task) in meditators. Jensen et al. [100,101] suggested the meditated group showed fewer errors in the commission of a Stroop task in contrast to the control group. Some research suggests that meditation training affects inhibition identified with verbal, as opposed to motor execution performance [102,103]. Conversely, other Stroop-based studies found no noteworthy performance differences among meditators and controls [104,105]. Nevertheless, Vega et al. [105] suggested that in the meditated group, a substantial error reduction was observed at follow-up which counteracted with longer response times, showing a speed–precision trade-off. Cahn and Polich [46] showed a decrease in the P3a ERP component, an orienting response to white noise bursts, following meditation. They interpreted this as reflecting improvement in cognitive control with mindfulness. Another study showed higher P300 amplitude at frontocentral brain regions in response to target stimuli (frontal P3b) following meditation practice, suggesting an improvement in frontal attention that was related to subsequent memory processing [41,106]. The current results for targets are in line with this latter study and extend understanding of the effects of mindfulness training on inhibitory aspects of the P300 (P3c); and more specifically show an increase in connection weights for P3c and frontal P3b. The weight of a connection reflects the amount of the signal input into the connection which passes between EEG channels and corresponds to the influence that the firing of one neuron has on another.

Other studies report reduced BOLD activation during mindfulness training [107,108]. Investigation of the precise relationship between connection weights, amplitude, and BOLD response warrants investigation and will be the focus of future research. Nevertheless, taken together, findings suggest that more efficient processing (reduced orientation to distractors, greater response to targets, increased inhibitory control) following mindfulness training might be underpinned by improvement in connectivity.

### 4.3. Pattern Classification (T1 vs. T2 vs. Post-Assessment)

In comparison to several conventional machine learning methods, the SNN models in the current study showed superior accuracy in classifying data from T1, T2, and T3. This may be due to several qualities inherent in the deSNN, compared to other models. With conventional machine learning techniques, the spatiotemporal features of every EEG sample are typically converted into one vector of features. Thus, the temporal connections cannot be obtained. These traditional methods can analyse the data in the form of static vectors and lack from exposing the spatiotemporal association. In comparison, the 3D brain-inspired SNN models allow the integration of dynamic interactions between underpinning brain functions during distinct psychological states. Moreover, rather than being a “black box”, deSNN allows visualisation of the changes in brain state as a function of group membership and experimental conditions.

SNN models can process time information alongside the spatial information while streaming the EEG data. Every training and testing sample includes the amplitude powers of all EEG channels within an entire time interval. Through the model training phase, the EEG temporal patterns are transferred into the SNN model via input neurons and the rest of the internal neurons in the model dynamically process these inputs. This is different from traditional machine learning techniques in which every data sample is presented as one data vector, where the integrated spatial and temporal information of the data is not preserved.

The brain-inspired SNN method is a generic architecture for precise analysis of various sorts of STBD (EEG and ERP). In comparison with conventional machine learning methods or deep learning techniques, SNN models have the following advantages: (1) they maintain both temporal and spatial components of the data in one model which is spatially regulated by a brain atlas; (2) they use a biologically plausible learning rule to detect spatiotemporal patterns from data; (3) they result in a better explanation of the spatiotemporal associations in the brain data variables. In this way, more accurate classification across assessment time points can be obtained.

### 4.4. Limitations and Future Research

For the current study, only EEG data of participants who had shown an increase in mindfulness scores (FFMQ) were analysed. Thus, no interpretation is made about the efficacy of mindfulness training, previously demonstrated in behavioural data. Furthermore, as with other ERP measures, the current scalp recorded data are limited in identifying precise cortical regions generating the activity. Given EEG data reflects activity from the superficial areas of the cortex, a more in-depth investigation of the fundamental brain regions needs to be undertaken.

The SNN models were trained on the whole EEG epoch for target and distractor stimuli (1300 ms post-stimulus). Future work should examine more closely smaller time windows around specific components. Although the present study has scrutinised the impacts of mindfulness training in response to target and distractors, our findings would not be limited to the defined stimuli. Therefore, further studies could examine whether similar impacts are detected in response to other types of stimuli (noise, background, etc.) followed by mindfulness training. Further studies are planned to model the deep patterns of connectivity learnt in the SNN models [70]. This will elicit an improved understanding of the individual effects of mindfulness for mental health and cognitive performance across population groups.

In summary, the designed brain-inspired SNN methodology for modelling brain data has provided several novel contributions to understanding ERPs. Current findings suggest that SNN models can provide useful information that differentiates brain states based on distinct task demands and stimuli, as well as changes in brain states as a function of psychological intervention. Thus, SNN-based methodology may provide a novel method for analysing EEG/ERP data that integrates temporal and scalp topographic information in understanding information processing.

## Figures and Tables

**Figure 1 sensors-20-07354-f001:**
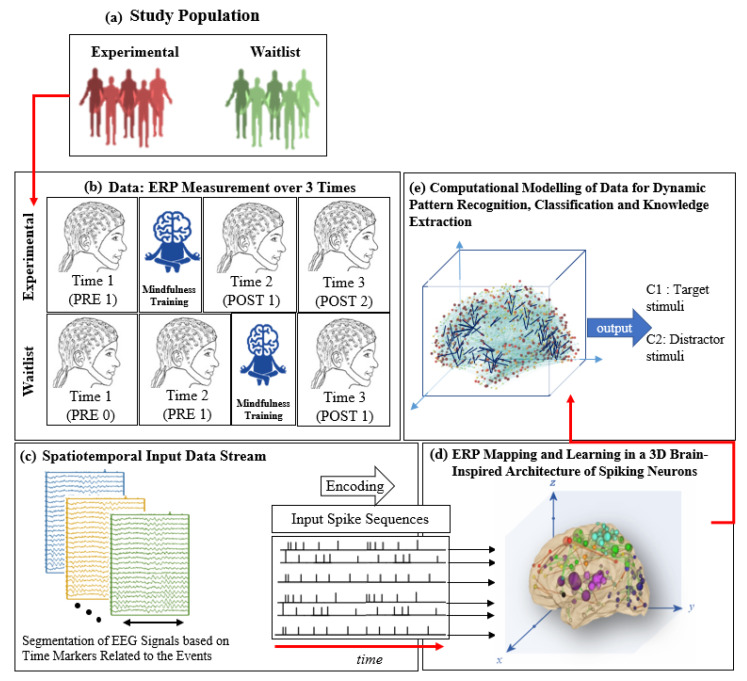
The diagram of event-related potential (ERP) data collection and computational modelling for two groups of participants. (**a**) Experimental and waitlist (control) groups. (**b**) Three times of ERP measurement (while performing an auditory cognitive task), before the training (baseline), after the training, and post-training. (**c**) Illustration of the designed methodology, containing segmentation of data based on the averaged ERP time points data post-stimulus extracted and ERP encoding into spike sequences. (**d**) Computational modelling of data into a 3D space of artificial neurons. (**e**) Pattern classification.

**Figure 2 sensors-20-07354-f002:**
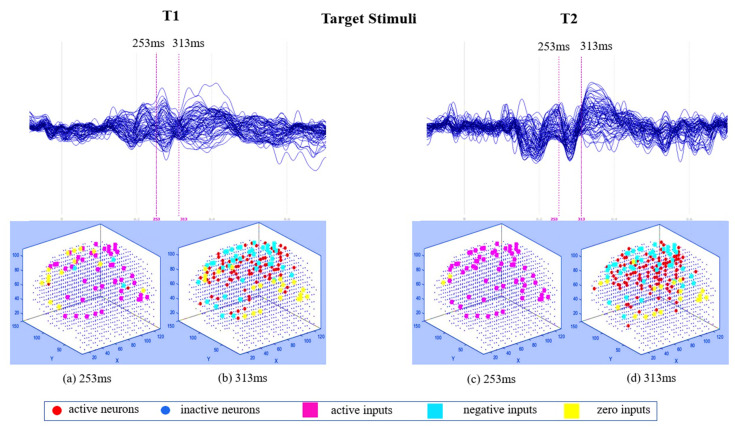
The averaged ERP waveforms across 62 EEG channels before (T1) and after (T2) the mindfulness training in response to target stimuli for an experimental participant. (**a**,**c**) Comparison of clusters of active inputs (positive spiking activity) in the SNN model corresponding to ERP latency of 253 ms post-stimulus at T1 and T2. (**b**,**d**) Comparison of clusters of negative inputs (negative spiking activity and the neuron firings) in the SNN model corresponding to ERP latency of 313 ms post-stimulus at T1 and T2.

**Figure 3 sensors-20-07354-f003:**
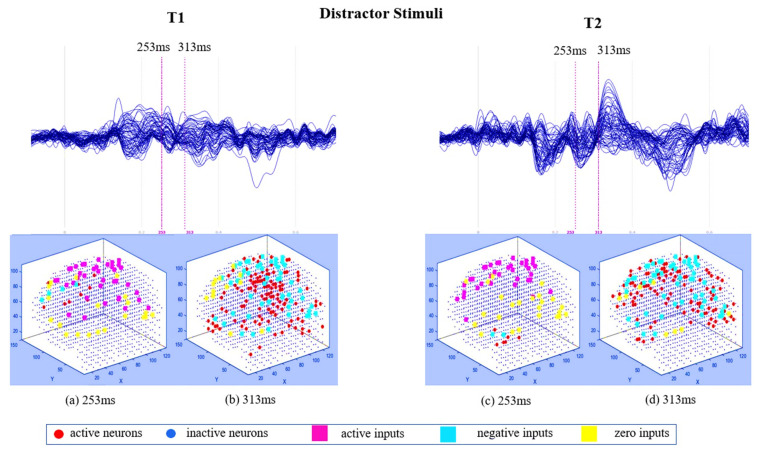
The averaged ERP waveforms across 62 EEG channels before (T1) and after (T2) the mindfulness training in response to distractor stimuli for an experimental participant. (**a**,**c**) Comparison of clusters of active inputs (positive spiking activity) in the SNN model corresponding to ERP latency of 253 ms post-stimulus at T1 and T2; (**b**,**d**) comparison of clusters of negative inputs (negative spiking activity and the neuron firings) in the SNN model corresponding to ERP latency of 313 ms post-stimulus at T1 and T2.

**Figure 4 sensors-20-07354-f004:**
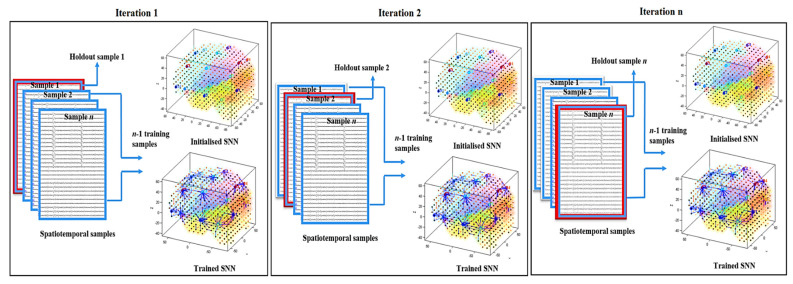
Iterative SNN modelling through the leave-one-out method. For *n* samples, the SNN model is initialised *n* times, and trained by a fold of different (*n* − 1) samples. Then the trained model is cross-validated by the hold-out sample.

**Figure 5 sensors-20-07354-f005:**
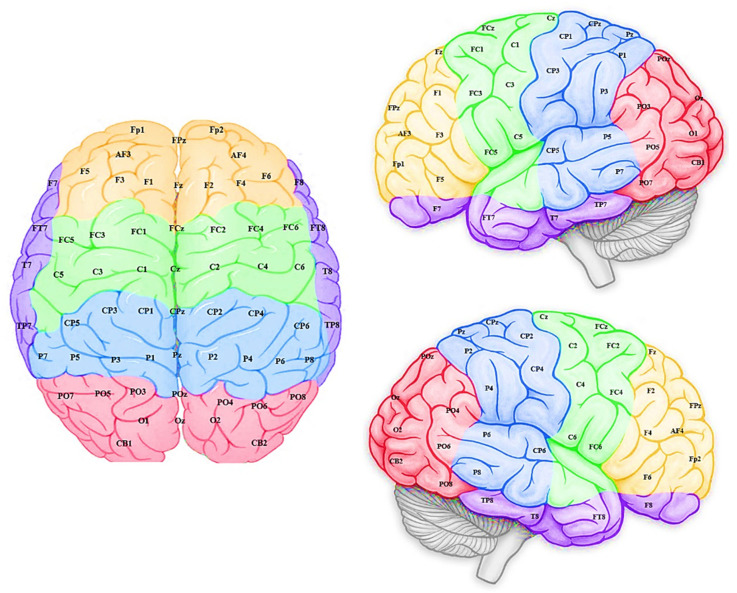
Dividing EEG channels into five sites for both hemisphere (left and right) with respect to their topological information including: yellow colour: left and right frontal (Fp1, AF3, F5,F3, F1 and Fp2, AF4, F6, F4, F2); green colour: left and right frontocentral (FC5, FC3, FC1, C5, C3, C1 and FC6, FC4, FC2, C6, C4, C2); purple colour: left and right temporal (F7, FT7, T7, TP7 and F8, FT8, T8, TP8); blue colour: left and right centroparietal (CP5, CP3, CP1, P7, P5, P3, P1 and CP6, CP4, CP2, P8, P6, P4, P2);pink colour: left and right occipitoparietal (PO7, PO5, PO3,O1, CB1 and PO8, PO6, PO4, O2, CB2).

**Figure 6 sensors-20-07354-f006:**
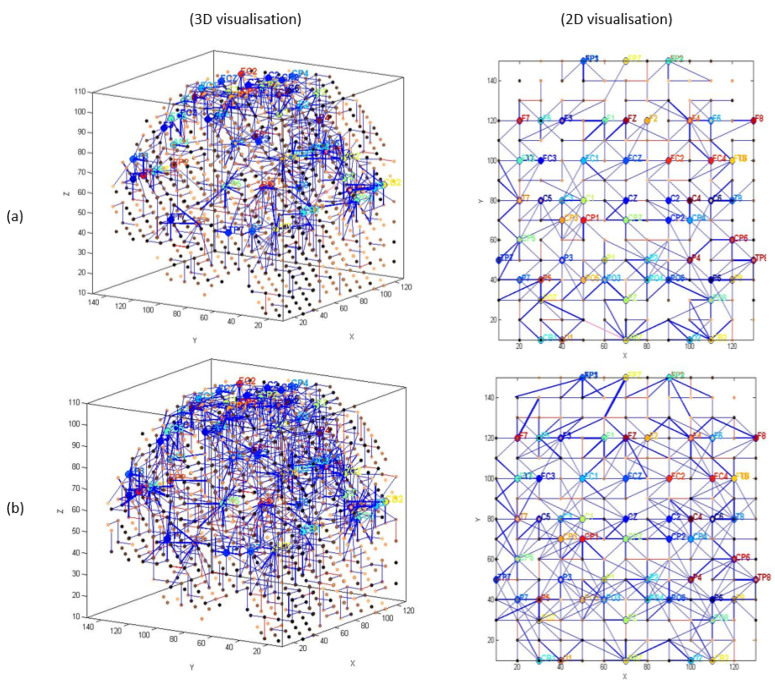
The neuronal connections created for the experimental group in the brain-inspired SNN models reflect the functional connectivity in response to target stimuli (**a**) before (T1) and (**b**) after the mindfulness training (T2).

**Figure 7 sensors-20-07354-f007:**
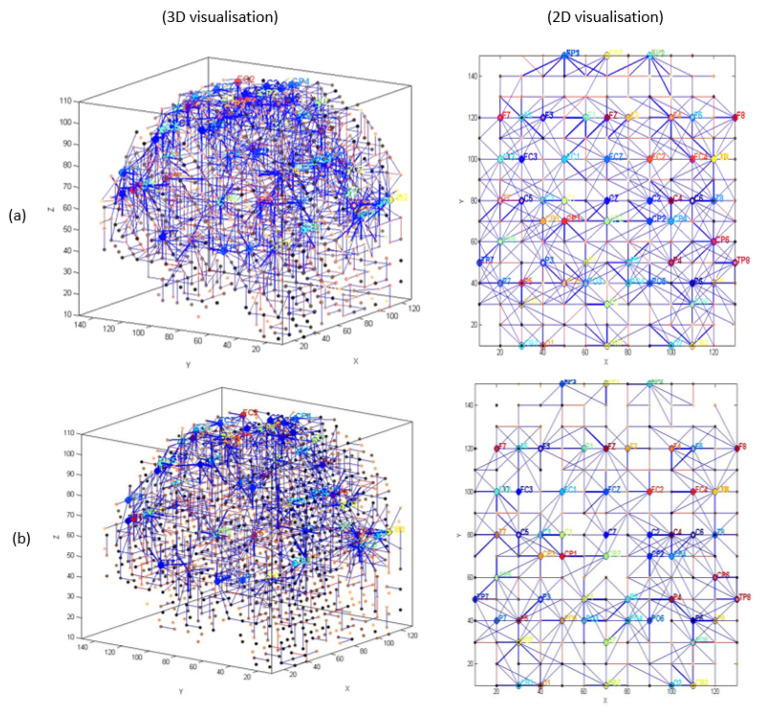
The neuronal connections created for the experimental group in the brain-inspired SNN models reflect the functional connectivity in response to distractor stimuli (**a**) before (T1) and (**b**) after the mindfulness training (T2).

**Figure 8 sensors-20-07354-f008:**
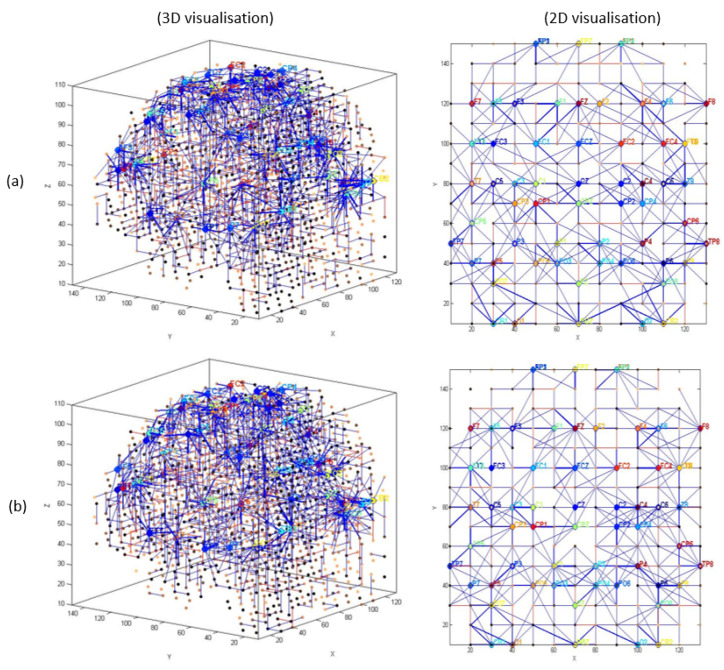
The neuronal connections created for the waitlist group in the brain-inspired SNN models reflect the functional connectivity in response to target stimuli (**a**) before (T1) and (**b**) after the mindfulness training (T2).

**Figure 9 sensors-20-07354-f009:**
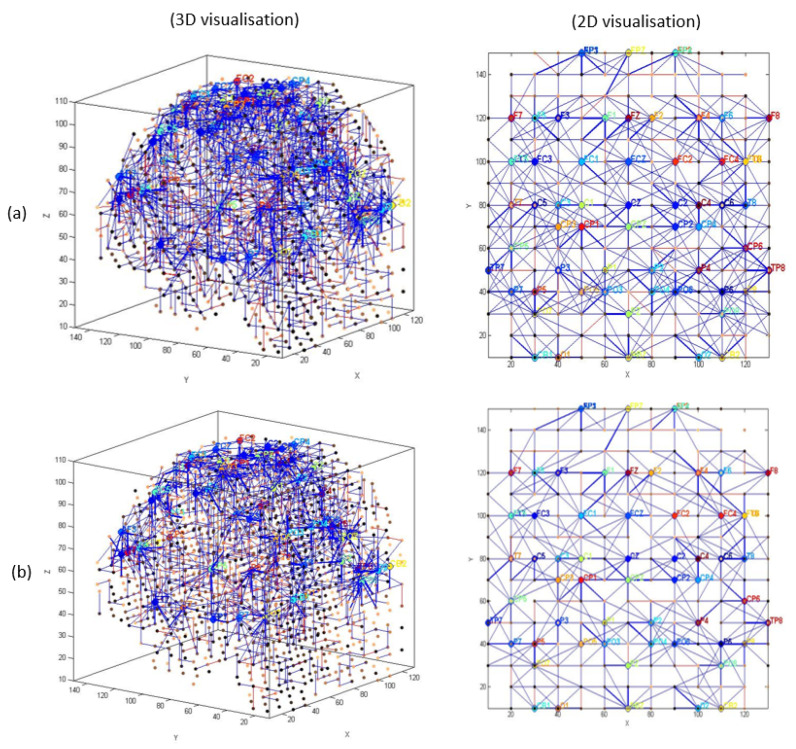
The neuronal connections created for the waitlist group in the brain-inspired SNN models reflect the functional connectivity in response to distractor stimuli (**a**) before (T1) and (**b**) after the mindfulness training (T2).

**Figure 10 sensors-20-07354-f010:**
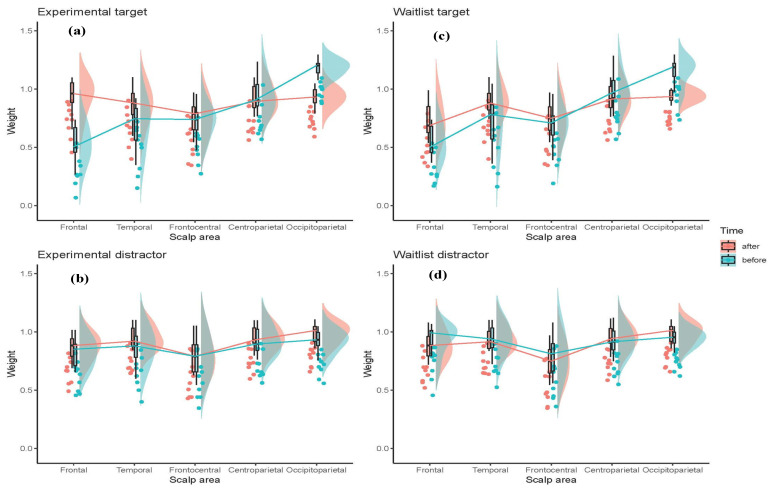
The SNN connection weights distribution related to “before” mindfulness (green colour) and six weekly mindfulness follow-ups (pink colour) across five sites of the brain (frontal, temporal, frontocentral, centroparietal, and occipitoparietal) in experimental and waitlist groups in response to (**a**,**c**) target stimuli and (**b**,**d**) distractor stimuli across the whole EEG epoch.

**Table 1 sensors-20-07354-t001:** The average connection weights measured for SNN models of target and distractor stimuli across five sites of the brain (frontal, temporal, frontocentral, centroparietal, and occipitoparietal) for both experimental and waitlist groups before training (T1) and after training (T2).

The Averaged Connection Weight of SNN Models					
Group	Site	Time Point			
		T1		T2	
		Type of Stimuli			
		Target	Distractor	Target	Distractor
Experimental	Frontal	0.53	0.82	0.94	0.87
	Temporal	0.69	0.87	0.87	0.96
	Frontocentral	0.72	0.80	0.77	0.81
	Centroparietal	0.95	0.92	0.92	0.95
	Occipitoparietal	1.18	0.92	0.93	0.99
Total weight		0.81	0.86	0.86	0.91
Waitlist	Frontal	0.55	0.94	0.73	0.87
	Temporal	0.74	0.94	0.88	0.95
	Frontocentral	0.7	0.81	0.75	0.85
	Centroparietal	0.99	0.92	0.92	0.95
	Occipitoparietal	1.15	0.95	0.94	0.99
Total weight		0.82	0.91	0.84	0.92

**Table 2 sensors-20-07354-t002:** Statistically significant ANOVA summary table for main effects and interactions of site weights. Significance =< 0.05. † = Degrees of freedom corrections applied using Greenhouse–Geisser due to violations of sphericity assumptions.

Variables	F-Value	Degrees of Freedom	*p*-Value	Partial-eta^2^ (η_p_^2^)
Site	18.432	4,72	<0.001	0.506
Time	24.468	1,18	<0.001	0.576
Stimuli	38.405	1,18	<0.001	0.681
Time * Group	14.962	1,18	0.001	0.454
Hemisphere * Stimuli	9.815	1,18	0.006	0.353
Site * Time †	44.631	2.761,49.702	<0.001	0.713
Site * Stimuli	40.842	2.135,38.434	<0.001	0.694
Time * Stimuli	5.786	1,18	0.027	0.243
Site * Time * Group	5.931	2.761,49.702	0.002	202

**Table 3 sensors-20-07354-t003:** Classification of 30 EEG samples (10 samples per participant) into three classes: target stimuli at T1-before mindfulness training (class 1), target stimuli at T2-after mindfulness training (class 2), and target stimuli at T3 (class 3) for experimental group and waitlist group. The classification method was leave-one-out-cross validation. The number of correctly classified samples in each class is located in the diagonal of the confusion table. The SNN-based classification accuracy was also compared with traditional methods: SVM, MLP, and MLR.

**SNN-Based Methodology (Experimental Group)**
**ERP Data Classes**	**Target Stimuli** **at T1(C1)**	**Target Stimuli** **at T2 (C2)**	**Target Stimuli** **at T3 (C3)**	**Accuracy** **(*%)***	**Total Accuracy (*%)***	**F-Score** **(%)**
Target Stimuli at T1(C1)	7	1	2	70.00	73.00	79.00
Target Stimuli at T2 (C2)	1	9	0	90.00
Target Stimuli at T3 (C3)	2	2	6	60.00
**Traditional Machine Learning Methods**
**Methods** **Accuracy in %**	**MLP** **50**	**SVM** **58.50**	**MLR** **48.50**	
**SNN-based methodology (waitlist Group)**
**ERP data Classes**	**Target Stimuli** **at T1(C1)**	**Target Stimuli** **at T2 (C2)**	**Target Stimuli at T3 (C3)**	**Accuracy (*%)***	**Total accuracy *%***	**F-Score** **(%)**
Target Stimuli at T1(C1)	7	2	1	70.00	73.00	79.00
Target Stimuli at T2 (C2)	1	7	2	70.00
Target Stimuli at T3 (C3)	1	1	8	80.00
**Traditional Machine Learning Methods**
MethodsAccuracy in %	MLP59.50	SVM51.50	MLR55.50

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
