# Peer review of "Interpretability of Spatiotemporal Dynamics of the Brain Processes Followed by Mindfulness Intervention in a Brain-Inspired Spiking Neural Network Architecture"

_sensors, 2020, doi:10.3390/s20247354_

Round 1

Reviewer 1 Report

Main comments:

This is an excellent paper combining neuroscience research with cutting edge AI learning.

I have the following suggestions for the authors to consider:

[1] I wonder if network analytics could be used to generate some statistics to summarize the properties of networks induced by the SNN models e.g. in Figures 7 to 9. Consider looking at Mark Newman or Albert Barabasi's works. Perhaps studying the properties of these interconnections could yield new insight.

[2] In particular, I wonder if [1] could help in understanding the relationship between site x group x time interactions in addition to the ANOVAs.

[3] in Table 3, traditional ML methods perform no better than chance while SNNs are appreciably better.

SNNs are extremely powerful. Of that I have no doubt. And so, I think the best test is to compete within itself. Consider scrambling the labels at least once and see if the SNNs still beat chance while the other traditional methods remain 50%?

Minor comments:

Please fix the missing citations on line 356.

Line 433 refers to Table 4. But there is no table 4 in the main manuscript.

Wilson

Reviewer 2 Report

This randomized placebo-controlled study aims at exploring spatiotemporal dynamics of brain processing indicators (ERPs) following mindfulness-based intervention. The study purpose was pursued using brain-spiking neural network (SNN) model. This is in my view one interesting and innovative study. It could contribute to the field through providing new information about those brain activation patterns that might be involved in mental imagery techniques, and spatiotemporal networks connectivity correlates of mindfulness interventions. The conceptualization of the study is very good and is supported by relevant literature sources. Overall, the manuscript is written in intelligible fashion. Still, some methodological issues appear in a must to be clarified. I have the following concerns.

  1. Cognitive Task to elicit ERP. It would be correct to call the employed paradigm a 4-stimulus oddball task, instead of “oddball” as it stands now. Next, it is recommended to point out the exact probabilities of different types (p = 62.5 vs. p = 12.5) in order to argue more directly the "oddball" nature of the paradigm. Also, it is necessary to argue the choice of this experimental paradigm both in the context of the objectives of the study and previous findings on the effects of meditation on ERP components as presented in the introduction. Providing the following explanations is clearly requested: 1) the choice of the paradigm and probabilities; 2) the expected information from each stimulus type, and from which aspects of early and late ERP correlates; 3) the role of the white noise stimulus; 4) the choice to use one and the same frequency for low probability targets and distractors, and for using the duration as a differentiating factor instead. To conclude, the functionality of the experimental paradigm should be argued with respect to the study outcome of interest.
  2.  EEG Acquisition and Pre-Processing. It is still not clear whether and how artifacts were detected and rejected reliably. The use of PCA here to reduce EOG/EMG contamination is questionable. This approach is in a must to be described and argued as much as thoroughly. The EOG and EMG registration set-ups also should be given in detail. What criteria were applied to classify EEG-related artifacts and to  consider epochs with EEG activity free of artifacts?  
